# The Use of Collagen-Glycosaminoglycan Biodegradable Matrix (Integra®) in the Management of Neck Postburn Hypertrophic Scars and Contractures

**Teodora Hoinoiu [1],[†], Daciana Grujic [2],[†], Georgeana Prilipceanu [3], Roxana Folescu [4],[†], Bogdan Hoinoiu [1],*, Tiberiu Bratu [2], Vladimir Poroch [5] and Ljubisa Grujic [3]**

[1]  Department of Clinical Practical Skills, Victor Babeş University of Medicine and Pharmacy, Timisoara, 300041 Timiș, Romania; tstoichitoiu@umft.ro

[2]  Department of Plastic, Reconstructive and Burn Surgery, Victor Babeş University of Medicine and Pharmacy, 2 Eftimie Murgu Square, Timisoara, 300041 Timiș, Romania; dcalistru@yahoo.com (D.G.); office@brol.ro (T.B.)

[3]  Clinic of Burns, Plastic and Reconstructive Surgery, "Pius Branzeu" Emergency County Hospital, 300723 Timisoara, Romania; gprilipceanu@gmail.com (G.P.); grujicmlj@yahoo.com (L.G.)

[4]  Department of Anatomy and Embryology, Victor Babeş University of Medicine and Pharmacy, 300041 Timişoara, Romania; roxanafolescu08@gmail.com

[5]  2nd Department of Internal Medicine, Grigore T. Popa University of Medicine and Pharmacy, 16 Universitatii Str., 700115 Iasi, Romania; vlader2000@yahoo.com

*  Correspondence: hoinoiu@umft.ro; Tel./Fax: +40-256-216510

†  These authors contributed equally to this work.

**Abstract:** Glycosaminoglycan (GAG) is a chain-like disaccharide that is linked to a polypeptide core to connect two collagen fibrils/fibers and provide the intermolecular force in a Collagen-GAG matrix which can be a valuable treatment of post-burn contractures and hypertrophic scars, which remain a challenge to reconstructive surgery. The face and neck contractures are the most difficult sites to treat. This article is meant to discuss our clinical experience in using collagen-glycosaminoglycan biodegradable matrix (Integra® Integra Lifesciences Corporation, Plainsboro, NJ, USA) to reconstruct defects created by excision of contracted areas from the neck and lower face areas. Between 2009 and 2011, we had 11 patients that underwent Integra reconstructive procedures. The mean follow-up period was 18 months. For all the patients, the intake of the Integra dermal regeneration template was 100%, even if one patient developed a minor infection treated with appropriate antibiotics. The patients are very satisfied with the result. A minor problem was a small difference in skin color, but this inconvenience was compensated by good skin elasticity.

**Keywords:** collagen-glycosaminoglycan biodegradable matrix; Integra dermal template; burns contracture; hypertrophic scars; keloids; surgical treatment

## 1. Introduction

Hypertrophic scars and keloids still represent a difficult management problem for plastic surgeons. Burns of the face and neck are a challenge for plastic surgeons because the lack of adequate coverage in this region can lead hypertrophic scars and keloids causing important functional limitations and subsequent disability. Scarring and tissue loss from second- and especially third-degree burns spontaneously healed or covered with skin grafts can usually cause deformities, and more rare are posttraumatic sequelae, or after the surgical excision of tumor [1–5]. Multiple treatment alternatives

have been used with variable but limited success (skin grafts, Z-plasties, local and distant flaps) [6–9]. During the last two decades, the development of Integra dermal regeneration templates associated with the use of skin graft and keratinocyte cultures has virtually solved these problems [10,11].

Since its introduction for the first time in 1981 by Burke et al. [12], collagen-glycosaminoglycan biodegradable matrix (Integra), has gained importance within our therapeutic arsenal for the coverage of dermal defects of diverse etiology. Integra is a system made of two layers: the deeper one is a matrix of bovine tendon collagen and shark glycosaminoglycan (condroitin 6 sulphate) with a porosity within 70–200 micrometers, and the superficial layer is a silicon sheet that prevents the fluid loss. The porosity of the deeper dermal substitute allows the anchoring of lymphocytes, macrophages, endothelium cells, and fibroblasts encouraging neovascularization and growth of so-called neodermis [13]. The upper silicone layers act like a temporary epidermis which seals the wound, and in this way prevent contamination and loss of heat and fluids. The neodermis grows in 3–4 weeks after the application of Integra; the upper silicone layer can be easily removed and replaced by an ultra-thin epidermal graft (the thickness recommended is between 0.05–0.15 mm). In spite of the fast healing of the donor sites, according to the literature, it is possible to use cultivated keratinocytes or composite biocompatible epidermal grafts [14–18].

Heimbach D.M reports the use of Integra in problematic cases such as acute burn treatment of the head and neck, which usually are difficult to reconstruct using conventional approaches that range from simple primary closure to microvascular free flaps. [19] Literature reports also the use of the dermal matrix for reconstruction of defects created by skin graft removal, excision of benign or cancer tumors and for covering of soft tissue following trauma. [20–23] Despite all the reports of burn wounds recovered do to the use of different artificial dermal templates, we find it relevant to share our experience in neck burn scars regarding reconstructive surgery with the applications of the Integra dermal regeneration template.

## 2. Material and Methods

In our clinic, we used collagen-glycosaminoglycan biodegradable matrix (Integra® Integra Lifesciences Corporation, Plainsboro, NJ, USA) for the first time in September 2009. Between September 2009 and November 2012, 11 patients ranging in age from 21 to 50 years (mean 34,3) with hypertrophic scars and retractions on the neck underwent reconstructive procedures with the Integra dermal regeneration template. Most of the patients were female (81.8%). The time between the burns and reconstruction ranged from one to eleven years (mean period was three years and five months). Most of the cases had burn scars localised on the anterior cervical region combined with retractile scars on the chin and of the body on the mandible (72.7%). More than half of patients underwent other reconstructive procedures before the Integra application. The average dimension of the remaining defect after scars excision was 140 mm/130 mm (Table 1).

The patients were admitted one day before surgery, and basic lab tests were performed. The patients were functionally evaluated before reconstruction with Integra analysing mentocervical angle. We consider mentocervical angle the angle formed by a line that is tangential to the submental point–from the chin to the subcervical region, and another tangential to the neck at the subcervical region intersection–the lowest point between the submental area and the neck. The scar tissue was washed before surgery with Betadine soap (Egis Pharmaceuticals PLC, Budapest, Hungary under licence MundipharmaAG, Basel, Switzerland) and 0.9% saline solution (Hemofarm A.D., Vršac, Srbija).

Under general anaesthesia, the surgical procedures consisted of two stages. In the first procedure, scars were excised until healthy tissue was well-vascularised. After excision, the defect was 25–30% higher than the initial scar size. Integra was rinsed three times in 0.9% saline solution and immersed for 10–15 min.

**Table 1.** Using Integra for the reconstruction of neck regions of eleven cases followed up.

| Case No. | Age (Years) | Sex | Time Accident-Integra (Years) | Scar Location | Other Reconstructive Procedures | Defect Size (mm) | Complications | Follow-up (Months) |
|---|---|---|---|---|---|---|---|---|
| 1. | 26 | F | 2 | Whole anterior and lateral cervical and inferior mandibular areas | Yes | 179/120 | None | 24 |
| 2. | 43 | F | 1 | Anterior and right laterocervical areas | No | 120/110 | Infection with *Staphylococcus aureus* | 12 |
| 3. | 21 | F | 10 | Anterior cervical and inferior mandibular | Yes | 150/115 | None | 16 |
| 4. | 50 | F | 11 | Anterior cervical, inferior mandibular and anterior thoracic areas | Yes | 120/140 | None | 18 |
| 5. | 32 | M | 1 + 2 months | Left laterocervical area | No | 160/150 | None | 14 |
| 6. | 28 | M | 2 | Anterior cervical, inferior mandibular and superior anterior thoracic areas | Yes | 110/100 | None | 24 |
| 7. | 34 | F | 3 | Left laterocervical area | Yes | 140/120 | None | 18 |
| 8. | 26 | F | 1 + 7 months | Anterior cervical, inferior mandibular and left genian areas | No | 120/140 | None | 12 |
| 9. | 43 | F | 1 + 8 months | Whole anterolateral area | No | 140/115 | None | 18 |
| 10. | 39 | F | 2 | Anterior cervical, inferior chin and anterior thoracic areas | Yes | 95/115 | None | 18 |
| 11. | 36 | F | 1 + 8 months | Right laterocervical, mandibular and right auricular areas | No | 90/125 | None | 24 |

After careful haemostasis, we applied Integra non-meshed tailored to fit the wound with the silicone part upside and we secured in place with staples. The most common option for dressing the wound was bolstered dressing, first applying a thick film of calendula ointment Epitelin (Aliphia, Timisoara Romania) as a non-adherent occlusive layer, on top Betadine soaked gauze and bandages to the edge [24]. The neck movements were restricted using a soft cervical collar. The next day, the wound was checked for any hematoma or serous collection. In the first week, the wound was inspected regularly every two days, and after that twice a week. Nine days after surgery, we saw the first signs of neodermis formation. Large spectrum antibiotics were being used for the first two weeks. Three weeks later, the second stage of the procedure was performed. The silicone layer was removed carefully with blunt instruments. The new dermal layer was debrided using a scalpel, excising the excess granular tissue, and then it was cleaned with a sterile brush. The remaining tissue was inspected in order to avoid any bleeding. The donor site for epidermal autograft was disinfected and infiltrated with a saline solution. The graft was harvested from posterior thorax with 0.10–0.15 mm thickness. After expansion, the graft was stapled and dressed with bulky bandages. The wound was dressed for ten days, and after that the patients started application moisturizing and healing ointments; Contractubex (Merz Pharmaceuticals GmbH, Frankfurt am Main, Germany), Cimeosil (Implantech Associates Inc, Ventura, CA, USA), Stratamed (Stratpharma AG, Basel, Switzerland) or any greasy ointment applied twice a day) for 6–9 months. Postoperative follow-up was 12–24 months.

Below we present 4 cases more meaningful from the 11 studied.

Case 1

A 26-year-old patient was admitted in our emergency burn unit with 58% burned surface, flame burns, 40% was IIIrd degree (face, neck, anterior and posterior thorax, upper limbs, both axillae, lower limbs). Eight months later, the patient presented to our clinic with contractures, keloids, and hypertrophic scars affecting the burned areas as well as the grafted parts, in spite of pressure garments and physiotherapy. First we considered the contracted scars from the limbs—we performed several interventions in order to

release the scars (from Z-plasty, local flaps and tissue expansion). The main problem was the face and the neck; the evolution of the scars was unsatisfactory. Two years after the burn trauma the neck scar made us consider Integra as an ideal treatment. (Figure 1A) The neck flexion contracture and hypertrophic scars were excised completely, and we applied Integra non-meshed and tailored to fit the wound, and nine days after surgery we saw the first signs of new neodermis formation (Integra becomes yellow with little red spots). (Figure 1B,C). Three weeks later we performed the second stage of the procedure. (Figure 1D). The patient was discharged seven days after surgery in good condition with the grafts completely integrated. Nine months after, the grafted area was soft, supple, pliable and elastic, without contracture and keloid formation, more alike to normal skin, the mobility of the head and neck was normal, and the patient was very satisfied with the result. (Figure 1E). The follow up of the patient at 11 years after the surgery showed the stability of the Integra matrix on the neck with the recovery of the pigmentation, and soft, good quality, and mobile skin without any sign of keloid (Figure 1F).

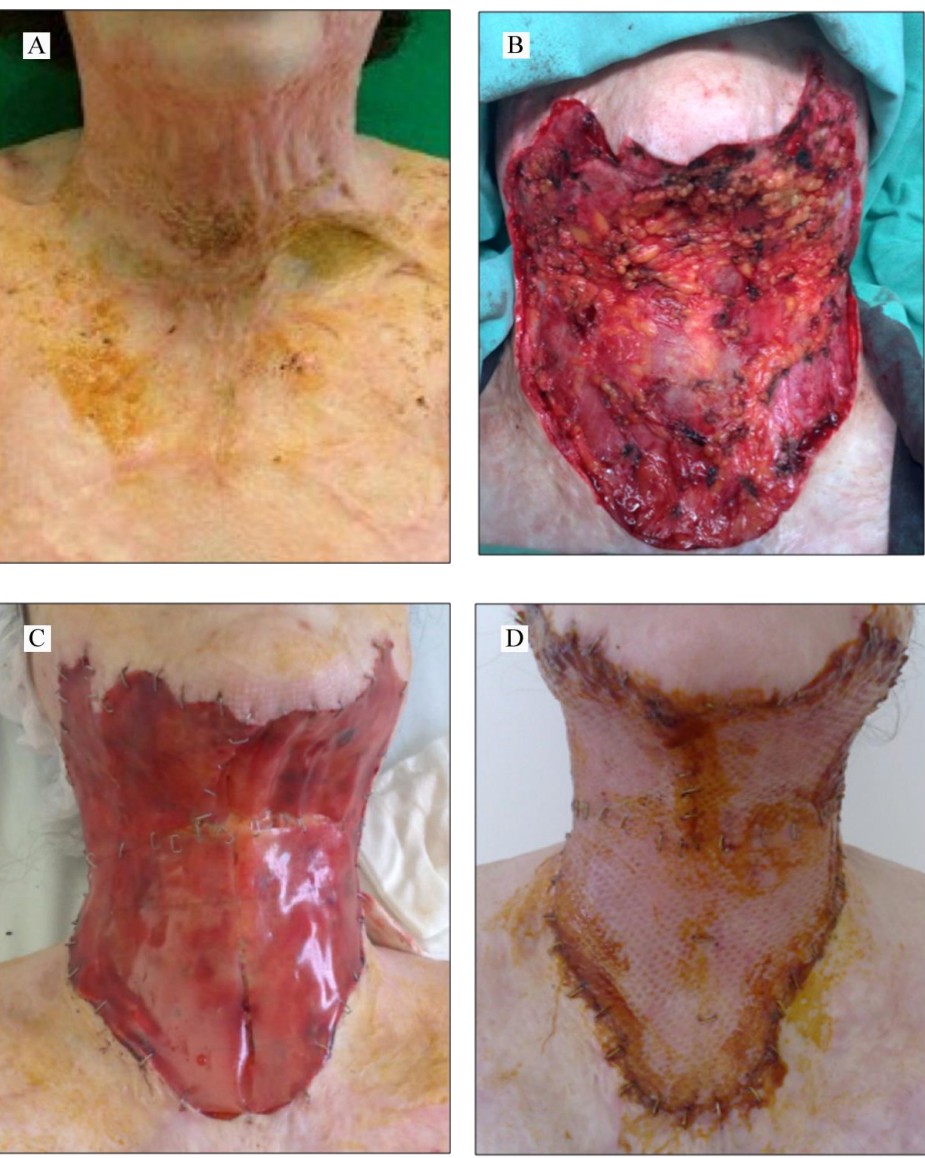

**Figure 1.** *Cont.*

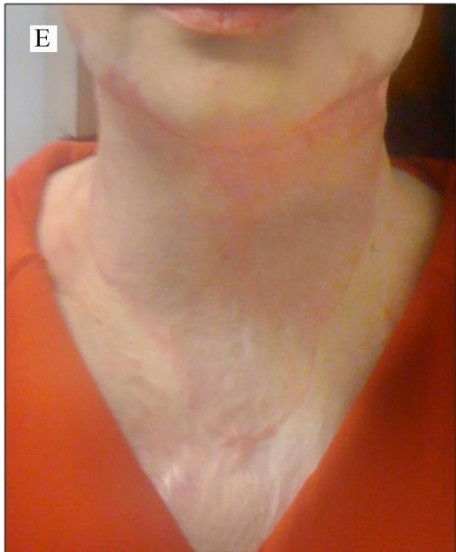 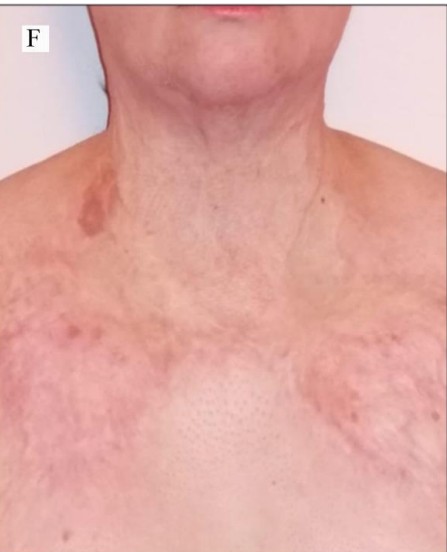

**Figure 1.** Case 1. A twenty-six years old female patient with 58% burned surface, flame burn (face, neck, anterior and posterior thorax, upper limbs, both axillae, lower limbs): (**A**) Aspect of the burn neck scar after two years-contractures, keloids and hypertrophic scars affecting the burned areas as well as the grafted parts; (**B**) image of the neck, after excision of the scars and contracted tissue; (**C**) final aspect of the neck seven days after application of epidermal autograft, frontal view; (**D**) final aspect of the neck seven days after application of epidermal autograft, frontal view; (**E**) Final aspect of the neck nine months after surgery. (**F**) Result at 11 years after surgery with soft, pliable skin more alike to normal skin.

Case 2

A 43-year-old female patient was admitted in our burn unit with 64% burned surface, chemical burns IIIrd degree (face, neck, anterior thorax, abdomen, upper and lower limbs). During three months, the scar was excised and grafted with autografts. One year after complete healing of burn lesions, the patient returned in our clinic with hypertrophic scars and contractures involving the right side of the neck and face, arms, axillae, right palm, abdomen and lower limbs. We decided to use Integra to improve contractures and scars for the right side of the neck and for the right palm. (Figure 2A) The scar tissue and contractures were excised and replaced with Integra (Figure 2B). Twenty days after surgery, the patient developed Staphylococcus aureus infection, treated for 5 days with Ciprofloxacin twice a day. One week later, the silicone layer was replaced with an epidermal autograft and after seven days, and the patient was discharged with healed grafted and donor areas (Figure 2C). Six months later, the skin was soft and easy to pinch, and the mobility of fingers was improved.

Case 3

A 21-year-old girl was admitted to our emergency burn unit with post-burn hypertrophic scars and contractures involving the face and the neck. The patient sustained scalds 10 years ago affecting the face, neck, and anterior trunk and hands about 25% body surface. She was treated by skin grafting of the IInd b and IIIrd degree burn areas. Three years later, she presented in our clinic with unsightly scars, contracture, and restricted neck movements (Figure 3A,B). We also chose Integra to improve contractures and scars. Nine months later, the skin was soft, easy to pinch, the skin colour was closer to normal and the neck movements were in the normal limits (Figure 3C,D).

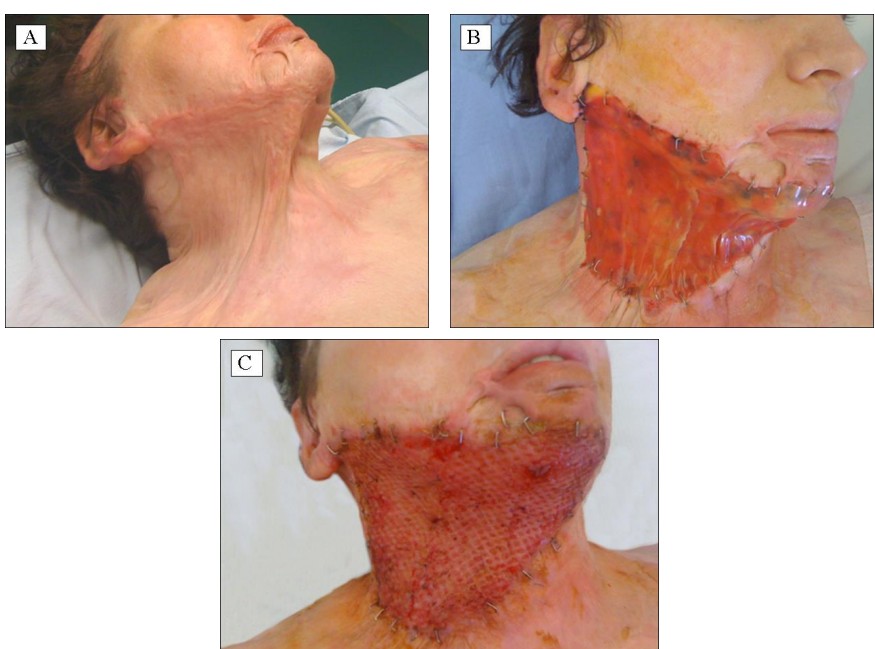

**Figure 2.** Case 2. A 43-year-old female patient with a 64% burned surface, IIIrd degree after chemical burn (face, neck, anterior thorax, abdomen, upper and lower limbs): (**A**) Hypertrophic scars and contractures involving the right side of the neck, one year after complete healing of burn lesions; (**B**) image of the neck skin, six days after reconstruction with Integra; (**C**) final aspect of the neck, seven days after the second stage of reconstruction (after covering with epidermal autograft).

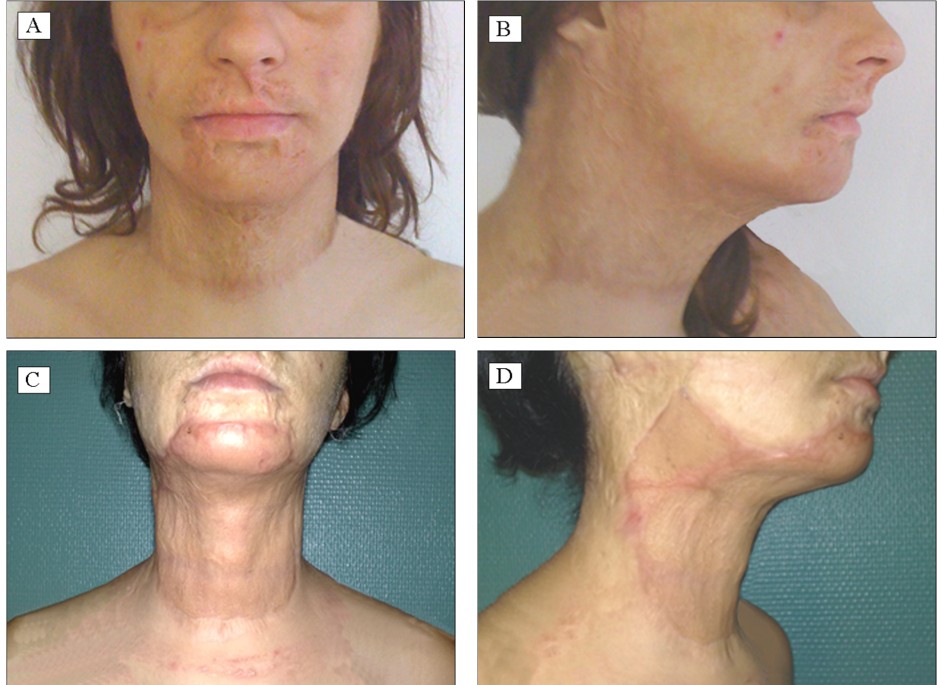

**Figure 3.** Case 3. A 21-year-old girl that suffered in her childhood, ten years ago a thermal burn by hot liquid IIndb–IIIrd degree on 25% burned surface (neck, anterior thorax, upper limbs): (**A**) Aspect of the neck scar prior to Integra reconstruction—frontal view; (**B**) aspect of the neck scar prior to Integra reconstruction—lateral view (**C**) neck skin reconstruction with Integra, 9 months later-frontal view (**D**) neck skin reconstruction with Integra, after 9 months—lateral view.

Case 4

A 50-year-old patient admitted with post-burn hypertrophic scars and contractures of the neck and anterior thorax, after repeated surgical procedures (skin grafting, tissue expansion) (Figure 4A).

The patient suffered eleven years ago a 30% body surface, flame burns, IInd–IIIrd degree on the cervical region, anterior thorax, upper limb. Using Integra was a good choice; (Figure 4B) the patient was very satisfied after 9 months (Figure 4C,D).

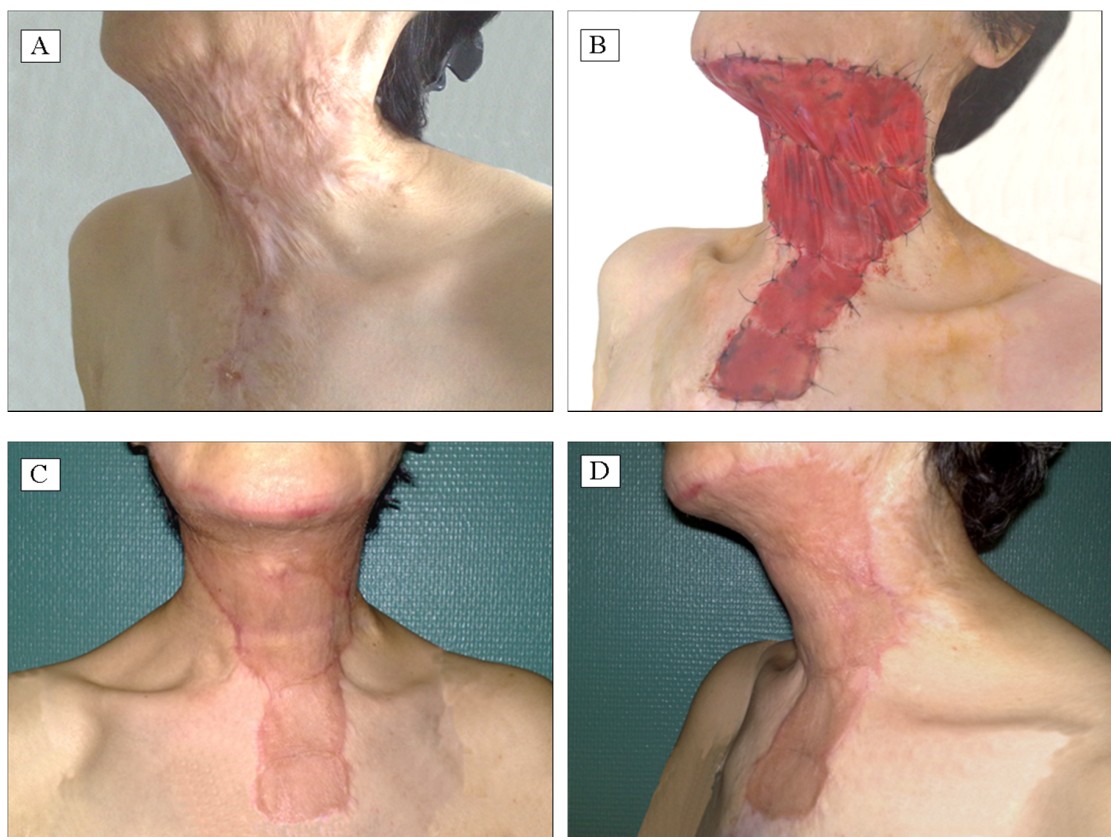

**Figure 4.** Case 4. A 50-year-old female patient suffered eleven years ago a 30% burned surface, flame burn, IInd-IIIrd degree (cervical region, anterior thorax, upper limbs): (**A**) Image of neck scar prior reconstruction with Integra; (**B**) image of the neck, 10 days after reconstruction with Integra; (**C**) skin neck reconstruction with Integra, after 3 months, frontal view; (**D**) skin neck reconstruction with Integra, after 3 months, lateral view.

## 3. Results

For all the 11 patients with post-burn neck contracture and hypertrophic scars, the intake of the Integra dermal regeneration template was 100%, even if one patient developed a minor Staphylococcus aureus infection treated with antibiotics for 5 days. The epidermal autograft showed no signs of infection or losses. The mean follow-up period was approximately 18 months.

We evaluated the aesthetics of the scars (vascularization, thickness, elasticity, and pigmentation) as well as functional results (Table 2). The functional outcome was evaluated after surgical procedures and functional recovery and relied mainly on the analysis of cervical extension angle value (mento-hyoido-sternal); this angle is under normal circumstances a maximum of 135°. An excellent result was considered if the cervical extension angle has a value of 115 ± 20°, a good result is considered when a cervical extension angle with a value of 90 ± 5°, and a poor outcome if a neck with an extension angle of less than 85° (Table 3).

The patients were very satisfied with the result. A minor problem was a small difference in skin color, but this inconvenience was compensated by the good skin elasticity and a good functional result. It was not necessary for subsequent reconstruction.

**Table 2.** Clinical evaluation of eleven cases after reconstruction with Integra, including local aspect and functional results.

| Patient No. | Age (Years) | Sex | Local Aspect | | | Functional Result | | |
|:---:|:---:|:---:|:---:|:---:|:---:|:---:|:---:|:---:|
| | | | Pigmentation | Vascularization | Elasticity | Excelent | Good | Poor |
| 1. | 26 | F | Normal | Normal | Very good | + | | |
| 2. | 43 | F | Normal | Normal | Very good | + | | |
| 3. | 21 | F | Normal | Normal | Very good | + | | |
| 4. | 50 | F | Normal | Normal | Very good | + | | |
| 5. | 32 | M | With pigmented areas | Normal | Good | | | + |
| 6. | 28 | M | With pigmented areas | Normal | Good | | | + |
| 7. | 34 | F | Normal | Normal | Very good | + | | |
| 8. | 26 | F | Normal | Normal | Very good | + | | |
| 9. | 43 | F | Normal | Normal | Very good | + | | |
| 10. | 39 | F | Normal | Normal | Very good | + | | |
| 11. | 36 | F | Normal | Normal | Very good | + | | |

**Table 3.** Evaluation of functional outcomes using the cervical neck angle (mento-hyoido-sternal angle) before and after reconstruction with Integra.

| Patient No. | Age (Years) | Sex | Cervical Extension Angle before Surgery | Cervical Extension Angle after Surgery | | |
|:---:|:---:|:---:|:---:|:---:|:---:|:---:|
| | | | | 95°–135° | 85°–95° | <85° |
| 1. | 26 | F | 64 | 129 | | |
| 2. | 43 | F | 79 | | 86 | |
| 3. | 21 | F | 85 | 115 | | |
| 4. | 50 | F | 73 | 114 | | |
| 5. | 32 | M | 80 | 109 | | |
| 6. | 28 | M | 75 | | 87 | |
| 7. | 34 | F | 74 | 108 | | |
| 8. | 26 | F | 80 | 108 | | |
| 9. | 43 | F | 81 | 103 | | |
| 10. | 39 | F | 69 | 98 | | |
| 11. | 36 | F | 83 | 108 | | |

## 4. Discussion

The treatment of post-burn contractures and hypertrophic scars remains a challenge for the reconstructive surgeon. These contractures should be treated as early as possible to prevent big functional deficits (daily activities, walking, eating, drinking) and of course, aesthetic impairment [25–27]. The face

and neck contractures are the most difficult sites to treat. All the techniques we can use sometimes give unsatisfactory results. Z-plasties, split-thickness skin grafts, and full-thickness skin grafts often heal with retraction and contractures. Local flaps, if they are available, can be bulky, cover only limited areas, and may necessitate revision surgery [28,29]. Because skin grafts do not bring a sufficient amount of dermis, skin dermis substituents were a good alternative. A type of these skin dermis substituents is Integra dermal template. Integra template was originally designed for immediate coverage of burn injuries, because a permanent dermis could rapidly be obtained [30,31]. It was noted, however, that the newly-created skin showed less or no tendency to hypertrophic scarring, keloid formation, or contracture, because no granulation or scar tissue was formed, resulting in superior function, cosmetics, and ultimately eliminating the need for scar treatment and pressure therapy. Anatomical sites such as the face and neck are likely to develop contraction and hypertrophic scar problems after conventional skin grafting; thus, Integra might be a reconstructive option. If in 1996 U.S. Food and Drug Administration (FDA) (within U.S. Department of Health and Human Services) approved the use of Integra for treatment of life-threatening burn injuries, it took 8 years for FDA to approve this device for the reconstruction of scar contractures. After 2006, there were several studies for using Integra for the coverage of tendons, joints, bones, and urinary bladder [32–34].

In the beginning, Heimbach et al. [29] used Integra in patients with major burns, and after that, its application has been expanded in reconstructive surgery for hypertrophic scars, keloids, and contractures. In 1990, Stern et al. [31] examined the histologic phases of wound healing in sites treated with Integra artificial skin and observed invasion of the Integra network by macrophages, lymphocytes, the formation of granulation tissue, neovascularization, and the formation of a neodermis with no adnexa or nerve endings. It was noticed that there was the minimal immune response to the bovine collagen and other components of Integra which did not influence the healing. Integra was used for reconstruction of defects created by excision of giant hair nevi, for the reconstruction of acute and chronic wounds, to protect exposed tendons, joints, and bones and to correct volume defects in rhinoplasty. In the rat models there are several studies to assess the feasibility of using this material in other sites: the bladder, the diaphragm, and the abdominal wall [35,36].

In this study, we used Integra for the treatment of post-burn hypertrophic scars, keloids, and contractures in the neck region, 1 to 11 years after the burn injury. We followed the classical recommended protocol. In the first week, we perform frequent dressings, every day, and after that we changed dressings twice a week. In the postoperative period we did not have any potential complications as hematomas, premature separation of the silicon layer, but we have one case of infection with *Staphylococcus aureus* treated with appropriate antibiotics based on wound cultures. The second stage of reconstruction did not raise any problems with epidermal autografts. The procedure was well-tolerated by all patients, and the results were almost comparable in softness, pliability, elasticity, and function to the normal skin (Table 2).

## 5. Conclusions

Integra dermal regeneration template reduces postoperative scar and contracture formation, has unlimited availability, is not bulky, and gives the same functional and cosmetic results as a full-thickness skin graft. It is a good choice for face and neck reconstruction and for all areas with limited tissue availability. The disadvantage is the cost of the product, which is expensive compared to other reconstruction procedures with autografts due to the surgical technique and the need for a good postoperative care, but the advantage stems from the overall cost of hospitalization [37].

**Author Contributions:** This research was made possible by the personal contributions of the following persons: T.H. for writing—original draft preparation, D.G. for conceptualization and methodology, G.P. performing the data curation, R.F. software support, B.H. writing—review and editing, T.B. for financial support, V.P. for methodology, L.G. for conceptualization. All authors have read and agreed to the published version of the manuscript.

**Funding:** Funding for this study was provided by grant from the Executive Agency for Higher Education, Research, Development, and Innovation Funding (Romania).

**Acknowledgments:** The authors thank all the members of the Clinic of Burns, Plastic and Reconstructive Surgery, "Pius Branzeu" Emergency County Hospital, Timisoara for supporting this study.

**Conflicts of Interest:** All the authors of this work wish to disclose there are no financial or other conflicts of interest that might bias the scientific information in the present article.

**Ethics Statement/Confirmation of Patient's Permission:** No ethical approval was required. We have obtained the patient's consent for publication of all images used.

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
