# Peer review of "The Use of Collagen-Glycosaminoglycan Biodegradable Matrix (Integra®) in the Management of Neck Postburn Hypertrophic Scars and Contractures"

_applsci, doi:10.3390/app10113731_

Round 1

Reviewer 1 Report

Notes to Authors

The Authors raise an extremely important aspect from the medical point of view regarding the surgical reconstruction of deformed skin surfaces. Since the nineties of the last century, intensive research began on the subject of transplantation, not only of burn skin, but also of the dura mater and spinal cord caused by craniocerebral traumas, meninges, meninges and spinal hernia.

Due to the nature of the topic discussed and the example of interesting Integra applications, I support the article for publication after taking into account the minor changes listed below.

  1. There are no literature reports regarding the use of Integra in other cases, no skin burns (not a solution or patent of Integra® Integra Lifesciences Corporation, Plainsboro, NJ, USA, not necessarily developed). Integra, i.e. a two-layer collagen biomaterial consisting of a silastic mesh layer that acts as an epidermis - preventing water loss and microbial infection, while on the other hand / a layer of highly porous structure consisting of bovine collagen and chondroitin 6-sulfate from shark cartilage. Literature reports on clinical transplants of the above biomaterial are known and developed already in the years 1990 - 2006, they concern, apart from application for skin tissue regeneration in patients with wounds or burn scars, also transplants in the case of replacing, supplementing or strengthening the dura mater and spinal cord in patients surgically treated for craniocerebral trauma, meningitis, meningitis, and for abdominal hernia repair.
  2. Are other interactions known (other than "not drying") the upper silicon / silastic layer of Integra with the growing tissues of the skin base? Can this topic be developed?
  3. Data regarding Integra should be included in the methodology (e.g. line 64 or 66) than in the abstract. Complete the description for Integra with: company, city, state.
  4. ”Integra analyzing cervical extension angle value (mento-hyoido-sternal angle)”- please refer to the description from another article, encyclopedia, etc. or explain more precisely what exactly the method consists of and what measuring devices are used to determine it, lines 75-76.
  5. Betadine soap - no producer / company, city, state, saline solution - no type of salt and its concentration (lines 76-77)
  6. Saline as above? at the same concentration? line 92
  7. Discussion of results:

    - Since the tests were performed at the turn of 2009-2012, it was possible to add photos of one or two patients from a later period, to confirm the effectiveness of Integra.;

    - Have you used creams, ointments for skin discoloration (based on compounds such as: caprylic / capric triglyceride, dicaprylyl ether, methylpropanediol, polyamide-5, tocopheryl acetate, glyceryl stearate etc.) for varied skin tone / pigmentation after transplantation? if not, it may be worth conducting such tests.;

    - It may also be worth considering what to do to make the price of Integra decrease enough to make it available for every patient, or possibly after a medical refund.

  8. Literature - minor errors like:
    - some links may and others do not, please harmonize it,
    - in some cases the range of pages cited is missing, e.g. citation number 3
    - parentheses, colons, semicolons are given, e.g. footnote No. 5, 13, please harmonize
    - no italics appear (e.g. on volumes).

Author Response

Dear Reviewer 1

We thank the reviewer for these insightful comments and for the special consideration that he gave to this manuscript and I respond to each of your question with paragraphs that I amended in the manuscript accordingly.

  1. There are no literature reports regarding the use of Integra in other cases, no skin burns (not a solution or patent of Integra® Integra Lifesciences Corporation, Plainsboro, NJ, USA, not necessarily developed). Integra, i.e. a two-layer collagen biomaterial consisting of a silastic mesh layer that acts as an epidermis - preventing water loss and microbial infection, while on the other hand / a layer of highly porous structure consisting of bovine collagen and chondroitin 6-sulfate from shark cartilage. Literature reports on clinical transplants of the above biomaterial are known and developed already in the years 1990 - 2006, they concern, apart from application for skin tissue regeneration in patients with wounds or burn scars, also transplants in the case of replacing, supplementing or strengthening the dura mater and spinal cord in patients surgically treated for craniocerebral trauma, meningitis, meningitis, and for abdominal hernia repair.

Heimbach D.M reports use of Integra in problematic cases such as acute burn treatment of head and neck, which usually are difficult to reconstruct using conventional approaches that range from simple primary closure to microvascular free flaps. Literature reports also the use of dermal matrix for reconstruction of defects created by skin graft removal, excision of benign or cancer tumors and for covering of soft tissue following trauma. Despite all the reports of burn wounds recovered do to use of different artificial dermal templates we find it relevant to share our experience in neck burn scars reconstructive surgery with the applications of Integra dermal regeneration template.

  1. Are other interactions known (other than "not drying") the upper silicon / silastic layer of Integra with the growing tissues of the skin base? Can this topic be developed?

The upper silicone layer act like a temporarily epidermis which seal the wound and in this way prevent contamination and loss of heat and fluids. The neodermis grows in 3-4 weeks after application of Integra the upper silicone layer can be easily removed and replaced by an ultra-thin epidermal graft (the thickness recommended is between 0.05-0.15 millimetres). In spite of the fast healing of the donor sites, according to the literature it is possible to use cultivated keratinocytes or composite biocompatible epidermal grafts.

  1. Data regarding Integra should be included in the methodology (e.g. line 64 or 66) than in the abstract. Complete the description for Integra with: company, city, state.

In our clinic we used collagen-glycosaminoglycan biodegradable matrix (Integra® Integra Lifesciences Corporation, Plainsboro, NJ, USA) for the first time in September 2009.

  1. ”Integra analyzing cervical extension angle value (mento-hyoido-sternal angle)”- please refer to the description from another article, encyclopedia, etc. or explain more precisely what exactly the method consists of and what measuring devices are used to determine it, lines 75-76.

We consider mentocervical angle the angle formed by a line that is tangential to the submental point- from the chin to the subcervical region and another tangential to the neck at the subcervical region intersection- the lowest point between the submental area and the neck.

  1. Betadine soap - no producer / company, city, state, saline solution - no type of salt and its concentration (lines 76-77)

The scar tissue was washed before surgery with Betadine soap (Egis Pharmaceuticals PLC, Budapest, Hungary under licence MundipharmaAG, Basel, Switzerland) and 0,9% saline solution (Hemofarm A.D., Vršac, Srbija).

  1. Saline as above? at the same concentration? line 92

Integra was rinsed three times in 0,9% saline solution and immersed for 10-15 minutes.

  1. Discussion of results:

- Since the tests were performed at the turn of 2009-2012, it was possible to add photos of one or two patients from a later period, to confirm the effectiveness of Integra.;

The follow up of the patient at 11 years after the surgery show the stability of the Integra matrix on the neck with recovery of the pigmentation, soft, good quality and mobile skin without any sign of keloid. (Fig. 1F).

- Have you used creams, ointments for skin discoloration (based on compounds such as: caprylic / capric triglyceride, dicaprylyl ether, methylpropanediol, polyamide-5, tocopheryl acetate, glyceryl stearate etc.) for varied skin tone / pigmentation after transplantation? if not, it may be worth conducting such tests.;

The most common option for dressing the wound was bolstered dressing, first applying a thick film of calendula ointment Epitelin (Aliphia, Timisoara Romania) as a non-adherent occlusive layer, on top Betadine soaked gauze and bandages to the edge.

The wound was dressed for ten days, and after that the patients started application moisturizing and healing ointments; Contractubex (Merz Pharmaceuticals GmbH, Frankfurt am Main, Germany), Cimeosil (Implantech Associates Inc, Ventura, USA) , Stratamed (Stratpharma AG, Basel, Switzerland) or any greasy ointment applied twice a day) for 6-9 months. Postoperative follow-up was 12-24 months.

- It may also be worth considering what to do to make the price of Integra decrease enough to make it available for every patient, or possibly after a medical refund.

The disadvantage is the cost of the product which is expensive and compared to other reconstruction procedure with autografts, the surgical technique and the need for a good postoperative care, but the advantage is coming from the overall cost of hospitalization.

  1. Literature - minor errors like:
    - some links may and others do not, please harmonize it,
    - in some cases the range of pages cited is missing, e.g. citation number 3
    - parentheses, colons, semicolons are given, e.g. footnote No. 5, 13, please harmonize
    - no italics appear (e.g. on volumes).

We will reconsider the minor error editing errors in order to achieve the standards of Applied Sciences Journal according to your recommendation.

Yours sincerely,

Bogdan Hoinoiu

Reviewer 2 Report

This is an interesting study about extending Integra applications to the difficult treatment sites of the face and neck.  I have one minor concern.  Please check Figure 1D.  Authors state in the legend that this image reflects the condition at nine months.  But the site appears to be in early stages of healing, with surgical staples still present.  Please correct the legend description of this image (shown in D) and also please provide the correct nine month image or better explain the unusual appearance.

Author Response

Dear reviewer

We are grateful for your positive evaluation and the insightful comments about our paper work. We will reconsider the minor text editing errors on the Figure 1D in order to achieve the standards of Applied Sciences according to your recommendation.

Yours sincerely,

Bogdan Hoinoiu